# The Progression of N6-methyladenosine Study and Its Role in Neuropsychiatric Disorders

**DOI:** 10.3390/ijms23115922

**Published:** 2022-05-25

**Authors:** Chunguang Lei, Qingzhong Wang

**Affiliations:** Institute of Chinese Materia Medica, Shanghai University of Traditional Chinese Medicine, Shanghai 201203, China; lcg2lt@gmail.com

**Keywords:** n6-methyladenosine, m6A-modifying enzymes, central nervous system functions, neuropsychiatric disorders

## Abstract

Epitranscriptomic modifications can affect every aspect of RNA biology, including stability, transport, splicing, and translation, participate in global intracellular mRNA metabolism, and regulate gene expression and a variety of biological processes. N6-methyladenosine (m6A) as the most prevalent modification contributes to normal embryonic brain development and memory formation. However, changes in the level of m6A modification and the expression of its related proteins cause abnormal nervous system functions, including brain tissue development retardation, axon regeneration disorders, memory changes, and neural stem cell renewal and differentiation disorders. Recent studies have revealed that m6A modification and its related proteins play key roles in the development of various neuropsychiatric disorders, such as depression, Alzheimer’s disease, and Parkinson’s disease. In this review, we summarize the research progresses of the m6A modification regulation mechanism in the central nervous system and discuss the effects of gene expression regulation mediated by m6A modification on the biological functions of the neuropsychiatric disorders, thereby providing some insight into new research targets and treatment directions for human diseases.

## 1. Introduction

Epigenetic modification refers to the reversible and heritable changes in gene function under the condition of an unchanged gene sequence and mainly includes DNA methylation and histone modification [1,2,3,4,5]. Similar to DNA epigenetics, RNA epigenetics also involves RNA chemical modifications that regulate gene expression by affecting RNA stability and translation [6]. To date, more than 150 types of RNA modifications have been identified, such as N1-methyladenosine [7], N7-methylguanosine [8], 5-methylcytosine [9], N6,2-dimethyladenosine glycosides [10,11], etc. These RNA modifications are widely distributed in various types of RNA, including messenger RNA (mRNA), transfer RNA (tRNA), ribosomal RNA (rRNA), small non-coding RNA, and long non-coding RNA (lncRNA) [12]. Among them, m6A modification is the most common chemical modification on mRNA discovered in recent years [13]. Studies have shown that m6A modification is involved in the metabolic activity of intracellular mRNA, regulates gene expression, and controls various biological processes such as RNA stability and mRNA translation [14]. Nevertheless, the detailed mechanisms of m6A modification have not been fully elucidated.

It has been reported that m6A modification is highly enriched in the brain and plays an important role in central nervous system development and neurodegenerative diseases [15]. The m6A modification is involved in the biological process of the central nervous system by regulating neural-related mRNA expression. When the activity or expression level of m6A-modifying enzymes in the brain is altered, the m6A modification level of related mRNAs is disrupted, leading to blockage of nuclear export, splicing, translation, and other processes, which in turn results in the occurrence of central nervous system diseases [16]. In this review, we describe the basic concept of m6A modification, the molecular mechanism of m6A modification in the regulation of RNA metabolism, and the biological role of m6A modification in the central nervous system and diseases, including a discussion of the clinical significance of m6A modification.

## 2. m6A RNA Methylation Mechanism

Desrosiers et al., 1975 first proposed a new RNA epigenetic modification, N6-methyladenosine (m6A), when analyzing the polyadenylic acid structure in a tumor cell, and found that approximately 80% of mRNA epitranscriptomic changes are m6A methylation modifications [17]. Since then, the presence of m6A has been detected in the RNAs of various eukaryotes and viruses such as yeast and Arabidopsis [18,19,20]. Recent studies have revealed that there are more than 7000 m6A-modified mRNAs in mammalian cells, and m6A also exists in ribosomal RNAs, transfer RNAs, small nucleolar RNAs, miRNAs, and long non-coding RNAs [12]. A comprehensive analysis of mRNA methylation has shown that the distribution of m6A in protein-coding regions, untranslated regions, and introns was 50.9%, 41.9%, and 2.0%, respectively. In the protein-coding region, m6A is mainly enriched near the stop codon, while in the untranslated region, m6A is mainly enriched at the 3’ end [11]. Analysis of the results of m6A high-throughput sequencing shows that m6A modification commonly occurs on the sequence of adenine of RRACH motif (R represents A or G, H represents U, A, or C) [10]. In mammals, the content of m6A modification varies in different tissues and is especially enriched in the liver, kidney, and brain [21]. In recent years, with the advancement of m6A modification detection methods, the mechanism of m6A modification has been gradually uncovered. The dynamic regulation mechanism of m6A modification mainly relies on three enzymes, namely methyltransferase complexes (“Writers”), demethylases (“Erasers”), and methyl readers (“Readers”) (Figure 1). Writers use S-adenosylmethionine (SAM) as a methyl donor to catalyze the formation of a methyl group at the sixth N element of adenine in RNA. At the same time, this modification can also be removed by an eraser to achieve dynamic adjustment. Readers are responsible for recognizing modification signals, which in turn affect the processes of RNA export, splicing, translation, and degradation. We will focus on the roles and functions of the three enzymes and the underlying mechanism of m6A modification.

### 2.1. m6A Methyltransferase (Writers)

The m6A-modified methyltransferase complex is mainly composed of METTL3, METTL14, WTAP, and KIAA1429 and also includes ZFP217 (Zinc Finger Protein 217), RBM15, RBM15B (RNA Binding Motif Protein 15B), HAKAI, ZC3H13, and other components [22,23]. Among them, METTL3 is the core component of methyltransferase that plays a catalytic role, and METTL14 is responsible for RNA recruitment, forms a heterodimer with METTL3, and jointly promotes the production of m6A modification. WTAP is responsible for stabilizing the complex, while RBM15 and RBM15B assist METTL3 in binding to WTAP so that they can precisely localize the target site and participate in the production of the m6A modification.

The m6A methyltransferase enzyme complex was first isolated with artificially synthesized RNA fragments from Hela cells, named MT-A (200 kDa) and MT-B (800 kDa), respectively [24,25]. Subsequently, MT-A1 and MT-A2 were successfully separated from MT-A. MT-A1 has a smaller molecular weight, only 30 kDa, and no methylase activity was found; MT-A2 as a multimer has a molecular weight of 200 kDa. The MT-A protein includes a 70 kDa protein with an S-adenosylmethionine binding site (MT-A70), which has the function of catalyzing m6A modification. MT-B has the largest molecular weight (875 kDa) in the enzyme complex, and has the properties of an RNA-binding protein, but its catalytic activity is low, and in the absence of MT-A1 and MT-A2, MT-B methylase activity disappeared [25]. The MT-A70 subunit contains two motifs, the S-adenosylmethionine (SAM) binding site and the DPPW motif (Asp-Pro-Pro-Trp) functional domain with catalytic function [25,26]. Immunofluorescence showed that METTL3 was mainly located in the nuclear speckles region in the nucleus, and this region was highly enriched for mRNA splicing factors, which indicated the potential role of METTL3 in the regulation of RNA metabolism [25].

METTL14 is an m6A-catalyzed methyltransferase discovered by homology analysis of the MT-A70 family [27]. Studies have shown that knockdown of METTL14 can reduce the extent of m6A modification in Hela cells and 293T cells, suggesting that METTL14 can catalyze the methylation reaction of m6A [28]; METTL3 and METTL14 can form a stable heterodimeric core complex with equal amounts, which functions as a methylase, and the complex of METTL3 and METTL14 has a strong affinity for the conserved sequence GGACU modified by m6A. METTL14 alone has higher m6A methylase activity than METTL3 alone, and when these two proteins combine into a heterodimer, the catalytic activity is significantly increased, and both are colocalized in the nuclear plaque region of the nucleus, playing a synergistic role in maintaining the stability of each other’s proteins. The downregulation of METTL3 or METTL4 leads to increased expression of certain target genes, suggesting that m6A may reduce mRNA stability [28]. Further investigation of the structure of the METTL3-METTL14 complex revealed that METTL3 has a catalytic subunit that serves mainly as the core of m6A’s catalytic activity, while METTL14 is mainly responsible for RNA recognition and binding while stabilizing the structure of the complex [22,29].

WTAP is the third identified m6A-modifying enzyme component. Originally, WTAP functions as a splicing factor for WT1 (Wilms tumor 1) protein and regulates cell cycle and embryonic development [30]. In Arabidopsis, the homologous protein mRNA adenosine methylase (MTA) of METTL3 was first found to interact with FIP37 (FKBP12 interacting protein 37) in vitro and in vivo, and FIP37 is the homologous protein of WTAP [31]. Meanwhile, Liu et al. (2014) found that WTAP can bind to the heterodimeric METTL3-METTL14 core complex and affect its catalytic activity of m6A modification [28]. Ping et al., 2014 found that WTAP interacts with the METTL3-METTLl4 complex, which may contribute to its localization in the pre-mRNA enriched nuclear plaque region and is essential for the catalytic activity of m6A. Under the absence of WTAP, the RNA-binding ability of METTL3 decreased significantly, suggesting that WTAP may regulate the recruitment of the m6A methyltransferase complex to mRNA targets [23]. METTL3 and METTL14 mainly bind the GGAC motif, WTAP mainly binds the GACU motif, and the three proteins share approximately 36% of the common binding motif [28].

RBM15 is another component of the m6A methyltransferase complex. Proteomic analysis revealed that RBM15, RBM15B, and WTAP may have a synergistic effect on m6A methyltransferase. The knockout of WTAP reduces the interaction between METTL3 and RBM15. Down-regulated expression of RBM15 and RBM15B significantly decreased m6A modification and reduced XIST-mediated downregulation [32].

### 2.2. m6A Demethylases (Erasers)

Similar to DNA and histone methylation, RNA has m6A demethylases that remove m6A-modified methyl groups. By 2011, FTO and ALKBH5 have been confirmed to be m6A-modified demethylating enzymes [33]. FTO was first discovered while studying fused toe mutations in mice [34], and subsequent studies found that the absence of FTO in mice leads to increased energy expenditure and systemic sympathetic activation. This leads to stunted growth and a marked reduction in adipose tissue and lean body mass [35]. In humans, mutations in FTO lead to a significant increase in body mass index, obesity, and a predisposition to diabetes [36]. Gerken et al., 2007 found that FTO belongs to the family of Fe(II) and 2-OG-dependent dioxygenases that catalyze the demethylation of single-stranded DNA [37]. In 2011, Jia et al., 2011 found that FTO can effectively demethylate m6A and that the knockout of FTO leads to an increase in the amount of m6A in mRNA, demonstrating for the first time the demethylase effect of FTO [33]; Xu et al., 2014 found that the expression of FTO significantly negatively correlated with the degree of m6A modification during adipogenesis. Downregulation of FTO can increase the level of m6A modification of the alternative splicing regulator SRSF2 (Serine and Arginine Rich Splicing Factor 2) and improve its RNA-binding ability. Additionally, FTO affects the exon splicing of the adipogenic regulator RUNX1T1 (Runx1 Partner Transcriptional Co-Repressor 1) by regulating the level of m6A near the splice site, thereby regulating adipose progenitor cell differentiation [38].

ALKBH5, the second m6A demethylase discovered in 2013, reverses the m6A modification of mRNA in vitro and in vivo. In HeLa cells, the downregulation of ALKBH5 increased m6A content in total mRNA by 9% and the overexpression of ALKBH5 decreased m6A content in total mRNA by 29% [39]. In addition, ALKBH5 can affect mRNA processing, nuclear export, and RNA metabolism through demethylase activity, and the level of mRNA m6A in mice with ALKBH5 gene deficiency is significantly increased, leading to apoptosis of spermatocytes in mid-meiosis and significantly weakening reproductive ability [39].

### 2.3. m6A Binding Proteins (Readers)

The main function of reader proteins is to recognize bases undergoing m6A modification and thus regulate the biological function of RNA. The identified m6A-binding proteins mainly include YTH domain-containing RNA-binding proteins, hnRNP, eIF3, IGF2BPs (insulin-like growth factor 2 mRNA-binding proteins), etc., which can specifically recognize and bind to m6A-modified mRNA and then play regulatory roles in mRNA stability, export, translation, splicing, and degradation [40,41,42,43,44].

YTHDF2 is the first m6A-binding protein to be discovered. Studies have shown that m6A can regulate the stability, processing, or translation of m6A-modified RNAs mainly by recruiting specific reader proteins or altering the RNA structure. YT521-B homology domain family (YTHDF) proteins are cytoplasmic recognizers of m6A that can specifically recognize and bind m6A-modified RRCH sequences and affect the stability and translation efficiency of RNAs containing m6A modifications; The direct interaction between the N-terminal region of CNOT1 and the SH region of the CNOT1 (Ccr4-Not Transcription Complex Subunit 1) subunit regulate target mRNA deadenylation and degradation [45,46].

YTHDF1 is another m6A-recognizing protein that was later identified. Investigating other functional roles of m6A modification, Wang et al., 2015 discovered another m6A-binding protein, YTHDF1, that can selectively recognize m6A-modified mRNA, promote its loading on ribosomes, enhance binding to translation initiation factor eIF3 and translation initiation, and affect gene expression by influencing mRNA translation and degradation [47].

It was also confirmed that YTHDF3 is an m6A-binding protein that promotes the translation of m6A-modified mRNA and may play a role in the initiation phase of translation via ribosomal proteins. The translation efficiency of the common targets of YTHDF1 and YTHDF3 was significantly enhanced, suggesting that the two may play a synergistic role in the translation process to produce their regulatory functions and that the effect of YTHDF3 on its own specific targets may be related to other RNA processing [48,49].

YTHDC1 is mainly located in the nucleus and can form YT bodies around the active transcription site and RNA processing structures. It was later found to bind specifically to m6A-modified RNA and regulate mRNA splicing; YTHDC1 and YTHDF2 share 21% of the m6A binding site. YTHDC1 has a stronger binding force to m6A modification and can easily bind to the GG (m6A) C sequence, and most of its binding sites are located around the stop codon [42]. Xiao et al., 2016 found that YTHDC1 can inhibit the binding of splicing factor SRSF10 (Serine and arginine rich splicing factor 10) to target mRNA by promoting the binding of SRSF3 (Serine and arginine rich splicing factor 3) mRNA, increasing its exons, and then regulating mRNA splicing [50].

The cytoplasmic m6A-binding protein YTHDC2 is critical for regulating m6A transcripts to ensure successful meiotic gene expression programs in mammalian reproduction [51]. Hsu et al., 2017 and Jain et al., 2018 found that YTHDC2 can specifically recognize m6A modification sites, promote translation of its target, and reduce its mRNA abundance [52,53].

The IGF2BPs family includes three RNA-binding proteins (RBPs), IGF2BP1 (Insulin Like Growth Factor 2 mRNA Binding Protein 1), IGF2BP2 (Insulin Like Growth Factor 2 mRNA Binding Protein 2), and IGF2BP3 (Insulin Like Growth Factor 2 mRNA Binding Protein 3) [54]. Huang et al. (2018) reported that the IGF2BPs family can recognize the modification site of m6A, function as an m6A-binding protein to bind to m6A-modified mRNA transcripts, and increase the stability of target mRNAs, such as MYC (Myc Proto-Oncogene, Bhlh Transcription Factor), thereby regulating gene expression [44].

The hnRNP family also contains three RNA-binding proteins, HNRNPC (Heterogeneous Nuclear Ribonucleoprotein C), HNRNPA2B1 (Heterogeneous nuclear ribonucleoprotein A2/B1), and HNRNPG (RNA binding motif protein X-linked). Among them, HNRNPC is an RNA-binding protein in the nucleus that is mainly responsible for processing pre-mRNA, m6A-modified mRNA, and lncRNA, and can alter their local structure to then regulate gene expression by regulating the alternative splicing of exons [55]. HNRNPA2B1, a member of the hnRNP family of RNA-binding proteins, has the ability to bind m6A-modified RNA substrates, and is involved in the regulation of biological processes such as miRNA precursor processing and alternative splicing of mRNA [56]. HNRNPA2B1 specifically recognizes the m6A site sequence RGACH (R = G/A, H = A/C/U), and then promotes the interaction with the pri-miRNA microprocessor complex protein DGCR8 (DGCR8 microprocessor complex subunit), promotes pri-miRNA maturation, and then regulates microRNA expression [56,57,58]. Subsequently, Wu et al., 2018 reported the crystal structures of HNRNPA2B1 in a complex with various RNA targets and elucidated the molecular basis of specific and multivariant recognitions of RNA substrates. The results of biochemistry and bioinformatics analysis suggested that m6A switches may be responsible for enhancing the accessibility of HNRNPA2B1 binding to specific binding sites [59]. Moreover, HNRNPG has a low-complexity AGG (Arg-Gly-Gly) structural domain that binds to RNAs with m6A modifications and regulates their stability and expression [60].

## 3. m6A Modification Is Involved in the Regulation of Central Nervous System Functions

The main function of the central nervous system (CNS) is to transmit, store, and process information, generate various psychological activities, and master and control all human behaviors. It has been found that m6A modifications are abundant in the central nervous system and have important effects on central nervous system functions, including brain tissue development, neural stem cell self-renewal and differentiation, neural synapse formation, the body’s ability to learn and remember, etc. [61,62,63,64].

### 3.1. m6A Modification and Brain Tissue Development

In the nervous system, as the brain gradually develops and matures, m6A modification also increases accordingly, suggesting that m6A modification is highly correlated with brain tissue development and maturation [11]. In order to study the effect of m6A modification on the development of the mouse cerebellum, after a specific knockout of METTL3 in the nervous system of mice, it has been found that the cerebellum was severely atrophied, and the weight of the whole brain and cerebellum was significantly reduced, and the decrease in m6A modification caused apoptosis in the cerebellum. A prolonged mRNA half-life of Dapk1 (Death Associated Protein Kinase 1), Fadd (Fas Associated Via Death Domain), and Ngfr (Nerve Growth Factor Receptor) eventually resulted in cerebellar hypoplasia [61]. The deletion of YTHDF2 may cause neural stem cells in the cerebral cortex of mice not to divide normally asymmetrically, resulting in the loss of neural progenitor cells, impaired differentiation of neurons, and a delay in the cortical development and dysfunction in the brains of mice [65]. In addition, ALKBH5 has been shown to be deficient in hypobaric and hypoxic environments, causing disturbance of the m6A modification of genes related to cerebellar development [66], further suggesting that m6A modification is closely related to brain tissue development.

### 3.2. m6A Modification and Neural Stem Cell Self-Renewal and Differentiation

Similarly, the self-renewal, proliferation, and differentiation of neural stem cells are also dependent on m6A modification [67,68]. Knockout of METTL14 in mouse embryonic neural stem cells can slow cell proliferation and cause premature differentiation. This process occurs mainly because the m6A modification sites of CBP/P300 (CREB binding protein) is reduced, increasing the stability of CBP/P300 mRNA, upregulating the expression of CBP/P300, promoting the acetylation of H3K27, upregulating differentiation-related genes, and inhibiting the expression of proliferation-related genes. The downregulation of METTL14 eventually results in blocking neural stem cell proliferation and advancing differentiation [62]. In adult neural stem cells, knockout of FTO in adult mouse stem cells can improve the proliferation and differentiation of neural stem cells in the short term but inhibits neurogenesis and neuronal differentiation in the long term. The deletion of FTO leads to a marked increase in m6A modification in Pdgfra (Platelet derived growth factor receptor alpha) and Socs5 (Suppressor Of Cytokine Signaling 5) and overactivation of the Pdgfra/Socs5-Stat3 pathway, resulting in blocked proliferation and differentiation of neural stem cells [69]. This was confirmed by another study, which showed that the absence of FTO caused a significant decrease in the number of adult stem cells in the subgranular zones (SGZ) of the hippocampus and the subventricular zone (SVZ) of the lateral ventricles, and also impairs the differentiation capacity of immature neurons [70].

### 3.3. m6A Modification and Neuronal Synapse Formation

It has been noted that the connection between synapses forms a complex neuronal network, and dysregulation of m6A modification impairs synapse formation and function [71]. Studies have shown that FTO is not only concentrated in the nucleus, but is also highly expressed in the axons of neurons [72]. Specific knockdown of FTO in axons increases m6A modification of GAP-43 (Growth associated protein 43) mRNA, which in turn reduces the local translation of GAP-43 mRNA and inhibits axons [72]. mRNA-seq and m6A-seq were utilized to organize the enriched m6A modification sites in the forebrain of mice to create a synaptic m6A epitranscriptome (SME). Subsequent experiments have shown that the loss of YTHDF1 or YTHDF3 leads to synaptic dysfunction and that the loss of YTHDF1 causes downregulation of the APC (adenomatous polyposis coli) gene in SME [63]. In addition, YTHDF1 and YTHDF2 are highly expressed in the axons of granulosa cells in the mouse cerebellum, and they have been shown to negatively regulate axon growth. The suppression of YTHDF1 or YTHDF2 can promote axon growth in granulosa cells. YTHDF1 activates the local translation of Dvl1 (Dishevelled Segment Polarity Protein 1) mRNA in axons, whereas YTHDF2 affects the local translation of Wnt5a (Wnt Family Member 5A) mRNA in axons by affecting the stability of Wnt5a mRNA, thereby synergistically negatively regulating axon growth [73]. In a word, the m6A methyl reader proteins have been found to have the function of regulating synaptic function and influencing nervous system stability.

### 3.4. m6A Modification Regulates Learning and Memory

The formation of long-term memory occupies an important position in learning and a variety of physiological processes, which are regulated by m6A modifications [74]. As for the relationship between m6A modification and the formation of memory, the ability of mice to consolidate memories is significantly decreased when METTL3 is knocked down in the mouse hippocampus. In contrast, the overexpression of METTL3 can consolidate memory and improves the learning and memory ability of the mice [75]. In the hippocampus of adult mice, YTHDF1 enhances the translation of hippocampus-associated mRNAs in response to external stimuli, which in turn promotes hippocampus-dependent spatial learning in mice and memory performance. When YTHDF1 in the hippocampus of adult mice is absent, neuroplasticity is blocked, leading to learning and memory deficits, and re-expression of YTHDF1 can repair the associated damage [74]. Coincidentally, a recent study also has shown that in Drosophila mushroom body neurons, the loss of METTL3 or YTHDF1/2/3 can induce short-term memory damage, and neither can compensate for the damage caused by the loss of the other, demonstrating that METTL3 and YTHDF1/2/3 are necessary for the promotion of normal short-term memory in mushroom body neurons [76]. In addition, FTO has been also reported to be expressed in the CA1 region of the dorsal hippocampus of mice, and negative feedback regulates the formation of memory, again showing that the formation of learning and memory depends on the regulation of m6A modification [77].

## 4. m6A Modification in Neuropsychiatric Disorders

Stable m6A modification contributes to the maintenance of central nervous system function. Conversely, disruption of m6A modification in the brain can lead to brain developmental delay and neuronal dysfunction. In central nervous system diseases, the disruption of m6A modification may be one of the most important reasons for the abnormal function of the nervous system leading to the occurrence of central nervous system diseases, including Glioblastoma, Alzheimer’s disease, Parkinson’s disease, depression, etc. [78,79,80,81].

### 4.1. m6A Modification and Glioblastoma

GBM (Glioblastoma) is a primary intracranial tumor and the most malignant glioma among astrocytic tumors. In a large number of glioblastoma samples, m6A-modified mRNA was found to be expressed in human glioma tissue, and the expression of METTL3 also showed a downward trend with the increase in malignancy grade. Later studies revealed that METTL3 can inhibit the proliferation, migration, and invasion of glioma cells and induce their apoptosis by inhibiting the protein phosphorylation level in the PI3K/Akt/mTOR pathway [82]. Cui et al., 2017 found that the downregulation of METTL3 and METTL14 decreased the mRNA methylation modification of ADAM19 (ADAM Metallopeptidase Domain 19) and increased the mRNA of ADAM19, while upregulating proto-oncogenes (such as ADAM19, EPHA3, and KLF4) and downregulating tumor suppressor genes (such as CDKN2A, BRCA2, and TP53I11). The astrocyte marker GFAP (Glial fibrillary acidic protein) and the TUBB3 (neuronal marker class III β-tubulin) were also downregulated, promoting self-renewal and tumorigenesis of GSCs [78]. Further treatment of GSCs with the FTO inhibitor MA2 revealed that their growth and self-renewal were significantly inhibited [78]. In addition, Han et al., 2021 found that METTL3 was downregulated in glioma tissue compared with normal brain tissue. The downregulation of METTL3 reduced the methylation of COL4A1 (Collagen type IV alpha 1) and increased its expression level, which promoted the proliferation and migration of glioma cells. Glioma stem cells (GSCs) are considered the initiating cells of glioblastoma, which can self-renew and differentiate in multiple directions and are an important cause of resistance to chemoradiation therapy and possible recurrence of gliomas [83]. Visvanathan et al., 2018 found that m6A RNA methylation of glioblastomas decreased during differentiation in vitro, possibly due to decreased expression of METTL3 during differentiation. The suppression of METTL3 in adenocarcinomas reduced the expression of adenocarcinoma-specific markers and significantly increased apoptosis. Further mechanistic studies revealed that in the presence of HuR (Human antigen R), METTL3 recognizes a specific site of the SOX2 (Sex-determining region of Y-box protein 2) mRNA 3′-UTR and performs m6A methylation modification to silence SOX2 mRNA, which mediates the stability of GSCs and the maintenance of stem cell properties [84].

Similar to METTL3, the expression of YTHDF2 was significantly lower in differentiated glioma cells than in undifferentiated GSCs [85]. Chai et al., 2021 have shown that the expression of YTHDF2 positively correlated with high malignancy grade, WHO glioma grade, and poor prognosis in gliomas. Biological mechanism studies revealed that YTHDF2 can promote the degradation of UBXN1 (UBX domain protein 1) mRNA by recognizing the m6A modification site mediated by METTL3. The degradation of UBXN1 mRNA and activation of the NF-κB pathway accelerated tumor progression [86]. Fang et al., 2021 found that YTHDF2 downregulates the expression of LXRα (Nuclear Receptor Subfamily 1 Group H Member 3) and HIVEP2 (Hivep Zinc Finger 3) genes through m6A-dependent mRNA decay [85]. Among them, LXRα maintains intracellular cholesterol homeostasis by regulating the uptake and excretion of cholesterol, which is essential for glioma proliferation and invasion [87], and HIVEP2 is a transcription factor whose downstream target SSTR2 can inhibit gliomas and also has a regulatory effect on MYC, NF-κB, and TGF-β signaling pathways [85]. Overall, the overexpression of YTHDF2 may simultaneously accelerate the degradation of UBXN1, LXRα, and HIVEP2 mRNAs, promoting glioma development through a complex network of actions. In glioblastoma stem cells, the m6A modification of MYC and VEGFA (Vascular endothelial growth factor A) was found to be upregulated. In contrast to the previously reported destabilization of mRNA by YTHDF2, in GSCs, YTHDF2 stabilized the transcription of mRNAs of MYC as well as VEGFA in an m6A-dependent manner, thereby promoting the expression of downstream IGFBP3 and accelerating the growth of GSCs. Then, the inhibition of the YTHDF2-MYC-IGFBP3 signaling pathway, thus inhibiting the growth of GSCs, may be a new way to treat GSCs [88]. In addition, some researchers found that ALKBH5 was highly expressed in glioma cells and that silencing ALKBH5 could inhibit the proliferation of glioma cells. Using transcriptome sequencing and m6A sequencing analysis, researchers found that ALKBH5 could de-methylate FOXM1 (Forkhead Box M1) and enhance its expression, which in turn promotes tumorigenesis [89].

### 4.2. m6A Modification and Alzheimer’s Disease

AD (Alzheimer’s disease) is a neurodegenerative disorder characterized by pathological manifestations of memory loss and cognitive impairment. For normal memory and learning function, METTL3 and METTL14 have been found to have an important relationship with long-term memory formation and normal striatal learning function in human and mouse hippocampi [75,90]. Huang et al., 2020 observed a high concentration of METTL3 in insoluble fractions in postmortem human AD samples, and the concentration correlated positively with the concentration of the insoluble tau protein. Therefore, the aberrant expression and distribution of METTL3 in the hippocampus of AD patients may represent the altered gene expression patterns associated with the pathogenesis of AD [91]. In the 3 × Tg-AD mouse model, overexpression of FTO can activate the target of rapamycin (mTOR) signaling and increase the phosphorylation rate of the neuronal tau protein. In contrast, knockout of FTO can inhibit mTOR signaling and decrease tau protein phosphorylation rather than tau mRNA and protein levels [92]. FTO has been shown to promote the occurrence of insulin defect-related AD by decreasing the TSC1 (TSC Complex Subunit 1) mRNA level, activating the mTOR signaling pathway, and promoting the phosphorylation of tau protein. Researchers speculated that FTO may demethylate the mRNA of TSC1 and reduce its stability, resulting in a decrease in its protein [93]. In addition, a recent study found that APP/PS1 transgenic mice, another mouse model for AD, had increased total methylation levels of m6A in the cortex and hippocampus compared with normal controls. At the same time, METTL3 is significantly increased and FTO is downregulated in APP/PS1 transgenic mice compared to normal mice [79]. Through high-throughput sequencing analysis, they also found that the methylation levels of m6A RNA of AMPA, NMDA, and SEMA genes encoding synaptic function in AD mice were different from those in the control group, and the methylation levels of AMPA and NMDA genes were increased, while the methylation levels of SEMA genes were significantly decreased [79]. In conclusion, differentially expressed m6A methylation plays an important role in the pathogenesis of AD.

### 4.3. m6A Modification and Parkinson’s Disease

PD (Parkinson’s disease), a neurodegenerative disorder characterized by tremors, bradykinesia, and muscle rigidity, is mainly due to the deformation and death of dopamine neurons in the substantia nigra of the midbrain, leading to a decrease in dopamine content in the striatum. Some researchers performed genome-wide association analyses (GWAS), differential gene analyses, and expression quantitative trait locus (eQTL) analyses to identify five m6A SNPs (rs75072999 of GAK, rs4924839, rs8071834, and rs1378602 of ALKBH5 and rs1033500 of C6orf10) associated with altered PD gene expression [94]. Studies have shown that the occurrence of PD is closely related to the m6A methylation modification, and its regulatory proteins play a crucial role in the apoptosis of dopaminergic neurons. In PC12 cells induced by the neurotoxin 6-hydroxydopamine (6-OHDA), a cellular model of PD, m6A methylation was reduced and FTO was highly expressed, whereas ALKBH5 showed no significant change [80]. In a PD rat model, m6A methylation levels were not significantly different in whole brain tissue, hippocampus, cortex, and midbrain, but were significantly reduced in the striatum region, and ALKBH5 expression was significantly increased, whereas there was no significant change in FTO. The overexpression of FTO in dopaminergic neurons reduces the level of mRNA m6A modification, induces the expression of ionotropic glutamate receptor 1 (N-methyl-D-aspartate receptor 1, NMDAR1), promotes oxidative stress and Ca2+ influx, and promotes degeneration or apoptosis of dopaminergic neurons [80,95]; while knockout of the FTO gene may have anti-apoptotic effects on dopaminergic neurons by reducing the expression of NMDAR1 [80]. Subsequently, they speculated that FTO leads to a decrease in m6A of NMDAR1 to ultimately stabilize NMDAR1 mRNA and thereby increase its expression. Thus, the m6A methylation modification affects the survival of dopaminergic neurons by regulating the expression of NMDAR1. In summary, m6A methylation modification plays an important role in the pathogenesis of PD, and FTO may be an effective drug target for the treatment of PD.

### 4.4. m6A Modification and Major Depressive Disorder

MDD (Major depressive disorder) is a psychiatric disorder clinically manifested by a loss of appetite, insomnia, depression, and, in severe cases, suicidal ideation, but its pathogenesis is still unclear. Some researchers genotyped 23 SNPs in the m6A modified genes from 1098 healthy individuals and 738 MDD patients. Among these SNPs, rs12936694 within ALKBH5 regions has been found to have a significant association with MDD [96]. Huang et al., 2020 performed high-throughput RNA sequencing in the hippocampus of mice treated with chronic unpredictable stress, and screened out circular RNA STAG1 (circSTAG1), microinjected circSTAG1 lentivirus into the hippocampus of mice, and observed the role of circSTAG1 in depression. The results showed that overexpressed circSTAG1 captured ALKBH5 and reduced the transport of ALKBH5 into the nucleus, promoting the increase in FAAH (Fatty acid amide hydrolase) mRNA m6A modification in astrocytes and the degradation of FAAH, alleviating astrocyte dysfunction and depression-like behavior caused by chronic unpredictable stress [97]. A recent study has shown that FTO was significantly downregulated in the serum of patients with major depression and in the hippocampus of mice with depressive-like behavior. Subsequently, they found that both FTO knockout mice and knockdown mice exhibited depression-like behaviors. This may be due to the increased methylation of Adrb2 (Adrenoceptor beta 2) mRNA caused by the deletion of FTO, which leads to a decrease in its mRNA and protein levels and is involved in the development of depression via the Adrb2-c-MYC-sirt1 pathway. After administration of the antidepressant fluoxetine, FTO was significantly upregulated, and depression-like behaviors improved, providing further confirmation that FTO is involved in the development of depression [98]. Coincidentally, another study has also shown that chronic restraint stress induces the impairment of synaptic plasticity in the mouse hippocampus, along with decreased protein levels of FTO, p-CaMKII (Phospho-Calcium/Calmodulin Dependent Protein Kinase II Gamma), and p-CREB (Phospho-Camp Responsive Element Binding Protein 1) [99]. Thus, there is increasing evidence to support the hypothesis that m6A-related enzymes play an important role in the development and prognosis of depression.

## 5. m6A Modifying Enzyme Inhibitors: A Potential Therapeutic Tool

In contrast to DNA methylation and histone modification, studies on small-molecule compounds targeting m6A-modifying enzymes have just begun [96]. However, current research indicates that small-molecule inhibitors of m6A-modifying enzymes have great potential for the development of novel therapies for a variety of diseases, including cancer, AML (acute myeloid leukemia), neuropsychiatric disorders, etc.

At present, several effective FTO inhibitors are known, which mainly include Rhein, R-2-hydroxyglutarate (R-2HG), HIF prolyl hydroxylation enzyme 2 inhibitors (such as IOX3), FB23, MO-I-500, Meclofenamic acid (MA), and so on. Among them, Huang et al. (2015) found that MA used as a nonsteroidal anti-inflammatory drug can compete to bind with FTO but not ALKBH5 to regulate mRNA containing m6A using a high-throughput screening method of fluorescence polarization and can selectively inhibit the activity of FTO [100]. In addition, MA as an inhibitor of FTO has been shown to inhibit the growth and self-renewal of GSCs and counteract the progression of glioblastoma [78]. Entacapone, a potent COMT inhibitor, is commonly used to treat Parkinson’s disease in combination with levodopa [101]. In 2019, using a virtual screening method, Peng et al. found that entacapone can selectively inhibit the demethylation of the FTO protein. They determined the crystal structure of the complex of the FTO protein and entacapone (PDB: 6AK4) and found that entacapone occupied both the cofactor and substrate binding sites [102]. Subsequent in vitro experiments also confirmed that entacapone can directly bind and inhibit FTO activity. In a diet-induced obesity mouse model treatment with entacapone, FOXO1, as a direct substrate of FTO, can induce intrahepatic gluconeogenesis and adipose tissue thermogenesis, resulting in weight loss and a decrease in the fasting blood glucose concentration [102]. The small-molecule inhibitors FB23 and FB23-2 can directly bind to the protein of FTO and selectively inhibit the m6A demethylase activity of FTO. From the results of an in vitro experiment and a mouse transplantation model, FB23-2, as the inhibitor of FTO, can significantly reduce the proliferation of human AML cell lines and primary AML cells, promote their differentiation and apoptosis, and significantly inhibit the malignant progression of tumors [103,104]. A recent study has shown that MO-I-500 as an FTO inhibitor could significantly reduce the negative effects of streptozotocin-damaged human astrocytoma cells CCF-STTG1 in the AD model [105]. In conclusion, FTO can be used as a drug target, and small-molecule inhibitors targeting FTO have the potential to treat various diseases.

Inhibitors targeting METTL3 received more attention as potential agents against cancer and other diseases. Bedi et al., 2020 used virtual screening methods and screened analogs and derivatives of SAM from more than 4000 compounds, of which six of them were selected for validation by protein crystallography, as well as two compounds with good ligand activity that can serve as potential METTL3 inhibitors [106]. Subsequently, another METTL3 inhibitor, UZH1a, is a small-molecule compound with strong binding to METTL3. UZH1a can induce apoptosis in AML-MOLM-13 cells by inhibiting the activity of METTL3 [107]. Meanwhile, the small molecule STM2457 was found to effectively bind to the active site of MEETL3 and inhibit its functional expression, resulting in decreased AML growth, increased differentiation, and apoptosis [108]. The above studies suggest that METTL3 inhibitors may be a potential treatment for AML.

In addition, there are relatively few studies on ALKBH5 inhibitors. Malakrida et al., 2020 found that imidazobenzoxazine-5-thione MV1035, as an ALKBH5 inhibitor, can decrease the migration and invasion of glioblastoma cell line U87, and showed that this change may be caused by decreased expression of the downstream protein CD73 [109]. This suggests that targeted inhibitors of m6A-modifying enzymes may provide a new avenue for disease therapy and also provide a potential mechanism of action for previously developed drugs.

## 6. Discussion

In recent years, with the rapid development of sequencing technology and the excavation of m6A methyl-modifying enzymes, the physiological modification function of m6A has been gradually revealed. As one of the most abundant modifications in cells, m6A is involved in biological processes such as cell division and differentiation, immune regulation, neurogenesis, and DNA damage by regulating RNA splicing, nuclear export, translation, and stability, leading to the occurrence of diseases. The disruption of m6A can lead to the occurrence of various diseases, including, but not limited to, cancer, leukemia, central nervous system diseases, etc. Therefore, maintaining the stable state of m6A modification in vivo by regulating the activity and expression of m6A modification enzymes may be a new avenue for new drug development, and a molecular compound targeted at the m6A modification enzyme may be used to develop potential drugs to treat these complex diseases.

A great deal of studies has suggested that alterations in the activity and expression of m6A methyl-modifying enzymes in the brain can affect brain development, the proliferation and differentiation of neural stem cells, the regeneration of neuronal synapses, and memory formation, and is then involved in the development of central nervous system diseases such as Alzheimer’s disease, Parkinson’s disease, glioblastoma, and depression (Figure 2). As Figure 2 shows, the development of CNS-related diseases may be mediated by multiple m6A enzymes responsible for conversion between methylated and unmethylated modification, which can be speculated by the dynamic process during transmission from healthy and disease states. Meanwhile, synthetic compounds and natural products will be discovered and used to treat central nervous system diseases by regulating the level of m6A modification in the brain in the future. In the end, the epitranscriptome of an m6A study will help to uncover the underlying pathophysiologic mechanism of neuropsychiatric disorders and provide an overview of the state-of-the-art drug discovery to treat human disease.

## Figures and Tables

**Figure 1 ijms-23-05922-f001:**
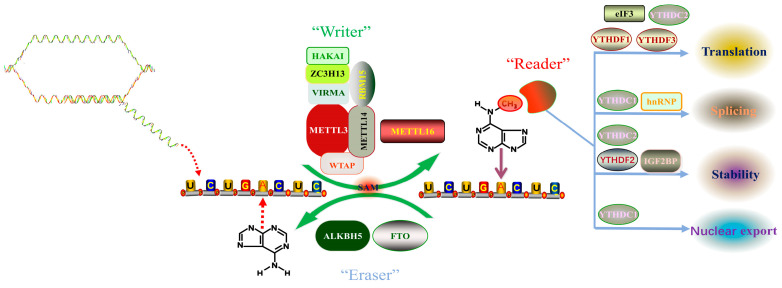
Dynamic m6A modification of RNA. Abbreviations: METTL3 (Methyltransferase-like protein 3); METTL14 (Methyltransferase-like protein 14); WTAP (Wt1 Associated Protein); KIAA1429 (Vir Like M6A Methyltransferase Associated, VIRMA); RBM15 (RNA Binding Motif Protein 15); HAKAI (Cbl Proto-Oncogene Like 1); ZC3H13 (Zinc Finger Ccch-Type Containing 13); METTL16 (Methyltransferase-like protein 16); FTO (Fat Mass and Obesity Associated Protein); ALKBH5 (Alkb Homolog 5, RNA Demethylase); eIF3 (eukaryotic initiation factor 3); hnRNP (heterogeneous nuclear ribonucleoprotein); YTHDF2 (Yth N6-Methyladenosine RNA Binding Protein 2); YTHDF1 (Yth N6-Methyladenosine RNA Binding Protein 1); YTHDF3 (Yth N6-Methyladenosine RNA Binding Protein 3); YTHDC1 (Yth Domain Containing 1); YTHDC2 (Yth Domain Containing 2).

**Figure 2 ijms-23-05922-f002:**
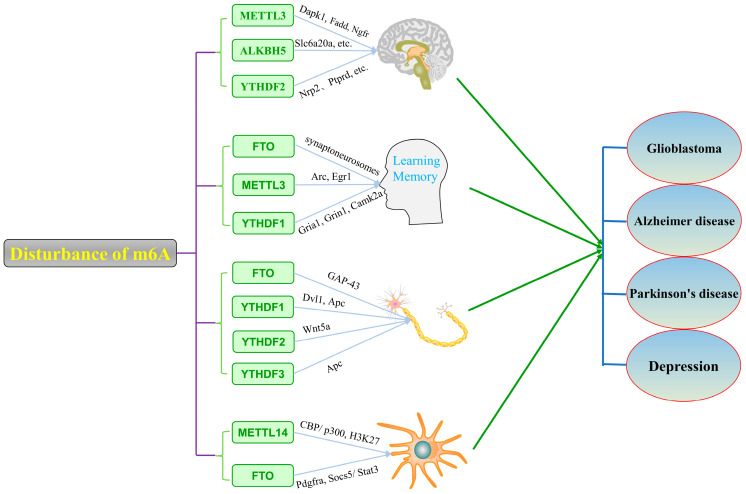
Disorders of m6A modification leads to dysfunction of the central nervous system and related diseases. Abbreviations: Slc6a20a (Solute carrier family 6 (neurotransmitter transporter), member 20A); Nrp2 (Neuropilin 2); Ptprd (Protein tyrosine phosphatase receptor type D); Arc (Activity regulated cytoskeleton associated protein); Egr1 (Early growth response 1); Gria1 (Glutamate ionotropic receptor AMPA type subunit 1); Grin1 (Glutamate ionotropic receptor NMDA type subunit 1); Camk2a (Calcium/calmodulin dependent protein kinase II alpha).

## Data Availability

Not applicable.

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
