# Peer review of "The Progression of N6-methyladenosine Study and Its Role in Neuropsychiatric Disorders"

_ijms, 2022, doi:10.3390/ijms23115922_

Round 1
Reviewer 1 Report
The present manuscript, focusing on the role of m6A in nervous system patho-physiology, is an excellent example of compendium on this issue.
The manuscript is clear and well-written.
I suggest to deal with some implementation.
- The authors should discuss also the role of non-canonical m6A reader reported in literature (for refs see also Wu 2018 nat comm, Alarcon 2015 cell).
- The authors should better indicate the genes whose methylation status can influence GBM, AD, PD and MDD.
- In the frame of the central role of RNA methylation in neuropsychiatric disorders, the authors should discuss the possible therapeutic approach to be applied to patients. For refs on the development of small molecules in therapy, see also Bedi 2020, Moroz-Omori 2021, Garbo 2021, Yankova 2021.
- Some typing error should be amended.
Author Response
Response to reviewer 1 comments
Point 1: The authors should discuss also the role of non-canonical m6A reader reported in literature (for refs see also Wu 2018 nat comm, Alarcon 2015 cell).
Response 1: Thank you very much for your valuable suggestions. We have added the studies on HNRNPA2B1 and HNRNPG in the lines 214-227 section of the m6A readers on page 5. The supplemented content has been listed as follows.
HNRNPA2B1, a member of the hnRNP family of RNA-binding proteins, has the ability to bind m6A-modified RNA substrates, and is involved in the regulation of biological processes such as miRNA precursor processing and alternative splicing of mRNA [56]. HNRNPA2B1 specifically recognizes the m6A site sequence RGACH (R=G/A, H=A/C/U), and then promotes the interaction with the pri-miRNA microprocessor complex protein DGCR8, promotes pri-miRNA maturation, and then regulates microRNA expression [56-58]. Subsequently, Wu et al. (2018) reported the crystal structures of HNRNPA2B1 in complex with various RNA targets and elucidated the molecular basis of specific and multivariant recognitions of RNA substrates. The results of biochemistry and bioinformatics analysis suggested that m6A switches may be responsible for enhancing accessibility of HNRNPA2B1 binding to specific binding sites [59]. Moreover, HNRNPG has a low-complexity AGG (Arg-Gly-Gly) structural domain of low complexity that binds to RNAs with m6A modifications and regulates their stability and expression [60].
Point 2: The authors should better indicate the genes whose methylation status can influence GBM, AD, PD and MDD.
Response 2: We appreciate this suggestion. According to the reviewer's comments, we added the methylation status and expression changes of downstream genes regulated by m6A methylating enzymes in the pharmacological studies of neuropsychiatric disorders.
The revised parts were listed as follows.
1) GBM parts (line of 317-364 in the page 7 and 8)
In a large number of glioblastoma samples, m6A-modified mRNA was found to be expressed in human glioma tissue and the expression of METTL3 also showed a downward trend with the increase in malignancy grade. Later studies revealed that METTL3 can inhibit the proliferation, migration, and invasion of glioma cells and induce their apoptosis by inhibiting the protein phosphorylation level in the PI3K/Akt/mTOR pathway [82]. Cui et al. (2017) found that downregulation of METTL3 and METTL14 decreased the mRNA methylation modification of ADAM19 and increased the mRNA of ADAM19, while upregulating proto-oncogenes (such as ADAM19, EPHA3, and KLF4) and downregulating tumor suppressor genes (such as CDKN2A, BRCA2, and TP53I11). The astrocyte marker glial fibrillary acidic protein (GFAP) and the neuronal marker class III β-tubulin (TUBB3) were also downregulated, promoting self-renewal and tumorigenesis of GSCs [78]. Further treatment of GSCs with the FTO inhibitor MA2 revealed that their growth and self-renewal were significantly inhibited [78]. In addition, Han et al. (2021) found that METTL3 was downregulated in glioma tissue compared with normal brain tissue. Downregulation of METTL3 reduced methylation of COL4A1 and increased its expression level, which promoted proliferation and migration of glioma cells. Glioma stem cells (GSCs) are considered the initiating cells of glioblastoma, which can self-renew and differentiate in multiple directions and are an important cause of resistance to chemoradiation therapy and possible recurrence of gliomas [83]. Visvanathan et al. (2018) found that m6A RNA methylation of glioblastomas decreased during differentiation in vitro, possibly due to decreased expression of METTL3 during differentiation. Suppression of METTL3 in adenocarcinomas reduced the expression of adenocarcinoma-specific markers and significantly increased apoptosis. Further mechanistic studies revealed that in the presence of human antigen R (HuR), METTL3 recognizes a specific site of the sex-determining region of Y-box protein 2 (SOX2) mRNA 3’-UTR and performs m6A methylation modification to silence SOX2 mRNA, which mediates the stability of GSCs and the maintenance of stem cell properties [84].
Similar to METTL3, the expression of YTHDF2 was significantly lower in differentiated glioma cells than in undifferentiated GSCs [85]. Chai et al. (2021) have shown that the expression of YTHDF2 positively correlated with high malignancy grade, WHO glioma grade, and poor prognosis in gliomas. Biological mechanism studies revealed that YTHDF2 can promote the degradation of UBXN1 mRNA by recognizing the m6A modification site mediated by METTL3. The degradation of UBXN1 mRNA and activation of NF-κB pathway accelerated tumor progression [86]. Fang et al. (2021) found that YTHDF2 downregulates the expression of LXRα and HIVEP2 genes through m6A-dependent mRNA decay [85]. Among them, LXRα maintains intracellular cholesterol homeostasis by regulating the uptake and excretion of cholesterol, which is essential for glioma proliferation and invasion [87], and HIVEP2 is a transcription factor whose downstream target SSTR2 can inhibit gliomas and also has a regulatory effect on MYC, NF-κB and TGF-β signaling pathways [85]. Overall, overexpression of YTHDF2 may simultaneously accelerate the degradation of UBXN1, LXRα, and HIVEP2 mRNAs, promoting glioma development through a complex network of actions. In glioblastoma stem cells, the m6A modification of MYC and VEGFA was found to be upregulated. In contrast to the previously reported destabilization of mRNA by YTHDF2, in GSCs, YTHDF2 stabilized the transcription of mRNAs of MYC as well as VEGFA in an m6A-dependent manner, thereby promoting the expression of downstream IGFBP3 and accelerating the growth of GSCs. Then, inhibition of YTHDF2-MYC-IGFBP3 signaling pathway and thus inhibiting the growth of GSCs may be a new way to treat GSCs [88].
2) AD parts (line of 374-386 and 390-394 in the page 8 and 9)
line of 374-386 in page 8
Huang et al. (2020) observed a high concentration of METTL3 in insoluble fractions in the postmortem human AD samples, and the concentration correlated positively with the concentration of insoluble tau protein. Therefore, the aberrant expression and distribution of METTL3 in the hippocampus of AD patients may represent the altered gene expression patterns associated with the pathogenesis of AD [91]. In the 3×Tg-AD mouse model, overexpression of FTO can activate target of rapamycin (mTOR) signaling and increase the phosphorylation rate of neuronal tau protein. In contrast, knockout of FTO can inhibit mTOR signaling and decrease tau protein phosphorylation rather than tau mRNA and protein levels [92]. FTO has been shown to promote the occurrence of insulin defects-related AD by decreasing the TSC1 mRNA level, activating the mTOR signaling pathway, and promoting the phosphorylation of tau protein. Researchers speculated that FTO may demethylate the mRNA of TSC1 and reduce its stability, resulting in a decrease in its protein [93].
line of 390-394 in page 8 and 9
Through high-throughput sequencing analysis, they also found that the methylation levels of m6A RNA of AMPA, NMDA, and SEMA genes encoding synaptic function in AD mice were different from those in the control group, and the methylation levels of AMPA and NMDA genes were increased, while the methylation levels of SEMA genes were significantly decreased [79].
3) PD parts (line of 404-422 in the page 9)
Studies have shown that the occurrence of PD is closely related to the m6A methylation modification and its regulatory proteins play a crucial role in apoptosis of dopaminergic neurons. In PC12 cells induced by the neurotoxin 6-hydroxydopamine (6-OHDA), a cellular model of PD, m6A methylation was reduced and FTO was highly expressed, whereas ALKBH5 showed no significant change [80]. In a PD rat model, m6A methylation levels were not significantly different in whole brain tissue, hippocampus, cortex, and midbrain, but were significantly reduced in the striatum region, and ALKBH5 expression was significantly increased, whereas there was no significant change in FTO. Overexpression of FTO in dopaminergic neurons reduces the level of mRNA m6A modification, induces the expression of ionotropic glutamate receptor 1 (N-methyl-D-aspartate receptor 1, NMDAR1), promotes oxidative stress and Ca2+ influx, and promotes degeneration or apoptosis of dopaminergic neurons [80, 95]; while knockout of the FTO gene may have anti-apoptotic effects on dopaminergic neurons by reducing the expression of NMDAR1 [80]. Subsequently, they speculated that FTO leads to a decrease in m6A of NMDAR1 to ultimately stabilize NMDAR1 mRNA and thereby increase its expression. Thus, the m6A methylation modification affects the survival of dopaminergic neurons by regulating the expression of NMDAR1. In summary, m6A methylation modification plays an important role in the pathogenesis of PD, and FTO may be an effective drug target for the treatment of PD.
4) MDD parts (line of 426-436 and 440-445 in the page 9 and 10)
line of 426-436 in page 9
Some researchers genotyped 23 SNPs in the m6A modified genes from 1098 healthy individuals and 738 MDD patients. Among these SNPs, rs12936694 within ALKBH5 regions has been found to be a significant association with MDD [96]. Huang et al. (2020) performed high-throughput RNA sequencing in the hippocampus of mice treated with chronic unpredictable stress, and screened out circular RNA STAG1 (circSTAG1), microinjected circSTAG1 lentivirus into the hippocampus of mice, and observed the role of circSTAG1 in depression. The results showed that overexpressed circSTAG1 captured ALKBH5 and reduced the transport of ALKBH5 into the nucleus, promoting the increase of fatty acid amide hydrolase (FAAH) mRNA m6A modification in astrocytes and the degradation of FAAH, alleviating astrocyte dysfunction and depression-like behavior caused by chronic unpredictable stress [97].
line of 440-445 in the page 10
This may be due to the increased methylation of Adrb2 mRNA caused by the deletion of FTO, which leads to a decrease in its mRNA and protein levels and is involved in the development of depression via the Adrb2-c-MYC-sirt1 pathway. After administration of the antidepressant fluoxetine, FTO was significantly upregulated and depression-like behaviors improved, providing further confirmation that FTO is involved in the development of depression [98].
Point 3: In the frame of the central role of RNA methylation in neuropsychiatric disorders, the authors should discuss the possible therapeutic approach to be applied to patients. For refs on the development of small molecules in therapy, see also Bedi 2020, Moroz-Omori 2021, Garbo 2021, Yankova 2021.
Response 3: We greatly appreciate this important suggestion. To present the progress of therapeutic studies, we have supplemented the section on drug discovery targeting m6A methylating enzymes, which includes the METTL3, FTO, and ALKBH5 pharmacological studies. The supplemented content has been listed as follows (lines 450-500 on pages 10 and 11).
5. m6A modifying enzyme inhibitors: a potential therapeutic tool
In contrast to DNA methylation and histone modification, studies on small-molecule compounds targeting m6A-modifying enzymes have just begun [96]. However, current research indicates that small molecule inhibitors of m6A-modifying enzymes have great potential for the development of novel therapies for a variety of diseases, including cancer, AML (acute myeloid leukemia), and neuropsychiatric disorders, etc.
At present, several effective FTO inhibitors are known, which mainly include, which mainly include Rhein, R-2-hydroxyglutarate (R-2HG), HIF prolyl hydroxylation enzyme 2 inhibitors (such as IOX3), FB23, MO-I-500, and Meclofenamic acid (MA), and so on. Among them, Huang et al. (2015) found that MA used as nonsteroidal anti-inflammatory drug can compete to bind with FTO but not ALKBH5 to regulate mRNA containing m6A using a high-throughput screening method of fluorescence polarization and can selectively inhibit the activity of FTO [100]. In addition, MA as an inhibitor of FTO has been shown to inhibit the growth and self-renewal of GSCs and counteract the progression of glioblastoma [78]. Entacapone, a potent COMT inhibitor, is commonly used to treat Parkinson's disease in combination with levodopa [101]. In 2019, using a virtual screening method, Peng et al. found that entacapone can selectively inhibit the demethylation of FTO protein. they determined the crystal structure of the complex of FTO protein and entacapone (PDB: 6AK4) and found that entacapone occupied both the cofactor and substrate binding sites [102]. Subsequent in vitro experiments also confirmed that entacapone can directly bind and inhibit FTO activity. In a diet-induced obesity mouse model treatment with entacapone, FOXO1 as a direct substrate of FTO, can induce intrahepatic gluconeogenesis and adipose tissue thermogenesis, resulting in weight loss and a decrease in fasting blood glucose concentration [102]. The small molecule inhibitors FB23 and FB23-2 can directly bind to the protein of FTO and selectively inhibit the m6A demethylase activity of FTO. From the results of in vitro experiment and mouse transplantation mode, FB23-2 as the inhibitor of FTO can significantly reduce the proliferation of human AML cell lines and primary AML cells, promote their differentiation and apoptosis, and significantly inhibit the malignant progression of tumors [103, 104]. A recent study has shown that MO-I-500 as an FTO inhibitor could significantly reduce the negative effects of streptozotocin-damaged human astrocytoma cells CCF-STTG1 in the AD model [105]. In conclusion, FTO can be used as a drug target and small molecule inhibitors targeting FTO have the potential to treat various diseases.
Inhibitors targeting METTL3 received more attention as potential agents against cancer and other diseases. Bedi et al. (2020) used virtual screening methods and screened analogs and derivatives of SAM from more than 4,000 compounds and 6 of them were selected for validation by protein crystallography and two compounds with good ligand activity that can serve as potential METTL3 inhibitors [106]. Subsequently, another METTL3 inhibitor, UZH1a, is a small molecule compound with strong binding to METTL3. UZH1a can induce apoptosis in AML-MOLM-13 cells by inhibiting the activity of METTL3 [107]. Meanwhile, the small molecule STM2457 was found to effectively bind to the active site of MEETL3 and inhibit its functional expression, resulting in decreased AML growth, increased differentiation and apoptosis [108]. The above studies suggest that METTL3 inhibitors may be a potential treatment for AML.
In addition, there are relatively few studies on ALKBH5 inhibitors. Malakrida et al. (2020) found that the imidazobenzoxazine-5-thione MV1035, as an ALKBH5 inhibitor, can decrease the migration and invasion of glioblastoma cell line U87, and showed that this change may be caused by decreased expression of the downstream protein CD73 [109]. This suggests that targeted inhibitors of m6A-modifying enzymes may provide a new avenue for disease therapy and also provide a potential mechanism of action for previously developed drugs.
Point 4: Some typing error should be amended.
Response 4: Thanks. We have corrected these errors.

Reviewer 2 Report
The manuscript named "The progression of N6-methyladenosine study and its role in the neuropsychiatric disorders" by the authors' Lei and Wang summarizes the role of the m6A epitranscriptome in the central nervous system. The manuscript is written very well. It gives the reader complex information regarding m6A modification and its biological functions, focusing on neuropsychiatric disorders.
After reading the manuscript, I have no major concerns regarding the presented work.
Some minor comments:
1) it would be nice if the authors provided a list of abbreviations
2) citation Desrosier et al., 1974 is missing in the reference list
3) manuscript should be checked for typing errors (such as WATP instead of WTAP etc.)
Overall I can recommend the manuscript for publication in its present form.
Author Response
Response to reviewer 2 comments
Point 1: It would be nice if the authors provided a list of abbreviations
Response 1: We greatly appreciated this suggestion. The abbreviations have been added to the table of abbreviations in this manuscript. The supplemented table was listed as follows.
Abbreviations
|
mRNA |
Messenger RNA |
|
tRNA |
Transfer RNA |
|
rRNA |
Ribosomal RNA |
|
lncRNA |
Long non-coding RNA |
|
SAM |
S-adenosylmethionine |
|
METTL3 |
Methyltransferase 3, N6-Adenosine-Methyltransferase Complex Catalytic Subunit |
|
METTL14 |
Methyltransferase 14, N6-Adenosine-Methyltransferase Subunit |
|
WTAP |
Wt1 Associated Protein |
|
KIAA1429 (VIRMA) |
Vir Like M6A Methyltransferase Associated |
|
ZFP217 |
Zinc Finger Protein 217 |
|
RBM15 |
RNA Binding Motif Protein 15 |
|
RBM15B |
RNA Binding Motif Protein 15B |
|
HAKAI (CBLL1) |
Cbl Proto-Oncogene Like 1 |
|
ZC3H13 |
Zinc Finger Ccch-Type Containing 13 |
|
MTA |
mRNA adenosine methylase |
|
FTO |
Fto Alpha-Ketoglutarate Dependent Dioxygenase |
|
ALKBH5 |
Alkb Homolog 5, RNA Demethylase |
|
SGZ |
Subgranular zones |
|
SVZ |
Subventricular zone |
|
GSCs |
Glioblastoma stem cells |
|
SRSF2 |
Serine And Arginine Rich Splicing Factor 2 |
|
RUNX1T1 |
Runx1 Partner Transcriptional Co-Repressor 1 |
|
hnRNP |
Heterogeneous Nuclear Ribonucleoprotein |
|
eIF3 |
Eukaryotic Initiation Factor 3 |
|
YTHDF2 |
Yth N6-Methyladenosine RNA Binding Protein 2 |
|
CNOT1 |
Ccr4-Not Transcription Complex Subunit 1 |
|
SME |
Synaptic m6A epitranscriptome |
|
YTHDF1 |
Yth N6-Methyladenosine RNA Binding Protein 1 |
|
YTHDF3 |
Yth N6-Methyladenosine RNA Binding Protein 3 |
|
YTHDC1 |
Yth Domain Containing 1 |
|
SRSF3 |
Serine And Arginine Rich Splicing Factor 3 |
|
YTHDC2 |
Yth Domain Containing 2 |
|
IGF2BPs |
Insulin-Like Growth Factor 2 mRNA-Binding Protein |
|
RBPs |
RNA-Binding Proteins |
|
IGF2BP1 |
Insulin Like Growth Factor 2 mRNA Binding Protein 1 |
|
IGF2BP2 |
Insulin Like Growth Factor 2 mRNA Binding Protein 2 |
|
IGF2BP3 |
Insulin Like Growth Factor 2 mRNA Binding Protein 3 |
|
MYC |
Myc Proto-Oncogene, Bhlh Transcription Factor |
|
HNRNPC |
Heterogeneous Nuclear Ribonucleoprotein C |
|
HNRNPA2B1 |
Heterogeneous nuclear ribonucleoprotein A2/B1 |
|
HNRNPG |
RNA binding motif protein X-linked |
|
CNS |
Central Nervous System |
|
AML |
Acute myeloid leukemia |
|
Dapk1 |
Death Associated Protein Kinase 1 |
|
Fadd |
Fas Associated Via Death Domain |
|
Ngfr |
Nerve Growth Factor Receptor |
|
Pdgfra |
Platelet derived growth factor receptor alpha |
|
ADAM19 |
A disintegrin and metallopeptidase domain 19 |
|
TUBB3 |
Class III β-tubulin |
|
GFAP |
Glial fibrillary acidic protein |
|
HuR |
Human antigen R |
|
SOX2 |
Sex-determining region Y-box protein 2 |
|
Socs5 |
Suppressor Of Cytokine Signaling 5 |
|
GAP-43 |
Growth Associated Protein 43 |
|
APC |
Adenomatous Polyposis Coli |
|
VEGFA |
Vascular endothelial growth factor A |
|
Dvl1 |
Dishevelled Segment Polarity Protein 1 |
|
Wnt5a |
Wnt Family Member 5A |
|
LXRα |
Nuclear Receptor Subfamily 1 Group H Member 3 |
|
HIVEP |
Hivep Zinc Finger 3 |
|
FOXM1 |
Forkhead Box M1 |
|
TSC1 |
TSC Complex Subunit 1 |
|
NMDAR1 |
Glutamate Ionotropic Receptor Nmda Type Subunit 1 |
|
FAAH |
Fatty Acid Amide Hydrolase |
|
p-CaMKII |
Phospho-Calcium/Calmodulin Dependent Protein Kinase II Gamma |
|
p-CREB |
Phospho-Camp Responsive Element Binding Protein 1 |
|
MA |
Meclofenamic acid |
Point 2: Citation Desrosier et al., 1974 is missing in the reference list.
Response 2: Thank you. This reference was added to page 2 at line 52-55 and to page 13 at lines 573-574, and the revised reference was listed as follows.
Desrosiers et al. (1975) first proposed a new RNA epigenetic modification, N6-methyladenosine (m6A), when analyzing the polyadenylic acid structure in tumor cell, and found that about 80% of mRNA epitranscriptomic changes are m6A methylation modifications [17].
Desrosiers, R. C.; Friderici, K. H.; Rottman, F. M. J. B., Characterization of Novikoff hepatoma mRNA methylation and heterogeneity in the methylated 5'terminus. 1975, 14, (20), 4367-4374.
Point 3: Manuscript should be checked for typing errors (such as WATP instead of WTAP etc.)
Response 3: Thanks. We have corrected these errors.

Round 2
Reviewer 1 Report
none